# Genome-wide proximity between RNA polymerase and DNA topoisomerase I supports transcription in *Streptococcus pneumoniae*

**María-José Ferrándiz[1], Pablo Hernández[2], Adela G. de la Campa** [1,3]*

**1** Unidad de Genética Bacteriana, Centro Nacional de Microbiología, Instituto de Salud Carlos III, Majadahonda, Madrid, Spain, **2** Centro de Investigaciones Biológicas Margarita Salas, Consejo Superior de Investigaciones Científicas (CSIC), Madrid, Spain, **3** Presidencia, Consejo Superior de Investigaciones Científicas (CSIC), Madrid, Spain

* agcampa@isciii.es

**Data Availability Statement:** The ChiP-seq data have been deposited in the NCBI Short Read Archive (SRA) database with the BioProject ID PRJNA656439. This data is available at https://

## Abstract

*Streptococcus pneumoniae* is a major cause of disease and death that develops resistance to multiple antibiotics. DNA topoisomerase I (TopoI) is a novel pneumococcal drug target. TopoI is the sole type-I pneumococcal topoisomerase that regulates supercoiling homeostasis in this bacterium. In this study, a direct *in vitro* interaction between TopoI and RNA polymerase (RNAP) was detected by surface plasmon resonance. To understand the interplay between transcription and supercoiling regulation *in vivo*, genome-wide association of RNAP and TopoI was studied by ChIP-Seq. RNAP and TopoI were enriched at the promoters of 435 and 356 genes, respectively. Higher levels of expression were consistently measured in those genes whose promoters recruit both RNAP and TopoI, in contrast with those enriched in only one of them. Both enzymes occupied a narrow region close to the ATG codon. In addition, RNAP displayed a regular distribution throughout the coding regions. Likewise, the summits of peaks called with MACS tool, mapped around the ATG codon in both cases. However, RNAP showed a broader distribution towards ATG-downstream positions. Remarkably, inhibition of RNAP with rifampicin prevented the localization of TopoI at promoters and, vice versa, inhibition of TopoI with seconeolitsine prevented the binding of RNAP to promoters. This indicates a functional interplay between RNAP and TopoI. To determine the molecular factors responsible for RNAP and TopoI co-recruitment, we looked for DNA sequence motifs. We identified a motif corresponding to a -10-extended promoter for TopoI and for RNAP. Furthermore, RNAP was preferentially recruited to genes co-directionally oriented with replication, while TopoI was more abundant in head-on genes. TopoI was located in the intergenic regions of divergent genes pairs, near the promoter of the head-on gene of the pair. These results suggest a role for TopoI in the formation/stability of the RNAP-DNA complex at the promoter and during transcript elongation.

www.ncbi.nlm.nih.gov/bioproject/PRJNA656439
The link to a UCSC genome browser session
displaying the uploaded sequence tracks is the
following: http://microbes.ucsc.edu/cgi-bin/
hgTracks?hgS_doOtherUser=submit&hgS_
otherUserName=Pablo_Hernandez&hgS_
otherUserSessionName=S_pneumoniae_ChIP-seq.

**Funding:** This work was supported by the
Ministerio de Economía y Competitividad
[BIO2017-82951-R to AGC]. The funders had no
role in study design, data collection and analysis,
decision to publish, or preparation of the
manuscript.

**Competing interests:** The authors have declared
that no competing interests exist.

## Author summary

*Streptococcus pneumoniae* is a main cause of pneumonia, meningitis and sepsis. Antibiotic
resistance in this bacterium has spread worldwide, compromising medical treatment.
Therefore, the development of new drugs directed to novel targets is necessary. DNA
topology is essential for the regulation of replication and gene expression. Topology is reg-
ulated and maintained by DNA topoisomerases, carrying out nicking-closing reactions.
Type I and type II topoisomerases act on single-stranded and double-stranded DNA,
respectively. Although type II topoisomerases are the target of clinically used antibiotics,
there are no clinical antibiotics directed against type I topoisomerases. Seconeolitsine, a
new drug targeting topoisomerase I, is effective against bacteria that have a single type I
topoisomerase, such as *Streptococcus pneumoniae* and *Mycobacterium tuberculosis*. In this
report, we studied the role of topoisomerase I in transcription. We found that topoisomer-
ase I and RNA polymerase physically interact *in vitro* and co-localize at gene promoters *in
vivo*. Binding of each of these enzymes to promoters was prevented by the specific inhibi-
tion of the other enzyme, supporting a role for topoisomerase I in RNA polymerase
transcription.

## Introduction

*Streptococcus pneumoniae* is the primary cause of community-acquired pneumonia, meningi-
tis, bacteremia and otitis media in children. In spite of the development of new vaccines, it
remains a serious cause of illness and death. Worldwide, one million children younger than 5
years old die annually of pneumococcal infections [1]. Given the spread of pneumococcal
resistance to beta-lactams and macrolides [2], nowadays treatment guidelines for pneumonia
recommend the use of fluoroquinolones [3]. However, although the prevalence of fluoroquin-
olones resistance is low, a rise in *S. pneumoniae* is predicted to occur as its use is increased. Flu-
oroquinolones target type II DNA topoisomerases: topoisomerase IV (TopoIV) and gyrase.
Topoisomerase I (TopoI) is another type I topoisomerase present in *S. pneumoniae*. These
three enzymes maintain DNA topology in this bacterium, solving topological problems associ-
ated with dynamic DNA remodeling. Finding new drug targets against *S. pneumoniae*, and
other pathogenic bacteria, is an urgent clinical need. TopoI has been proposed as a new anti-
bacterial target [4]. Our group has reported the use of a novel drug, seconeolitsine (SCN), to
inhibit both TopoI activity and growth of *S. pneumoniae* [5,6] and *Mycobacterium tuberculosis*
[7], without affecting human cell viability. SCN causes an increase of negative DNA-supercoil-
ing (Sc) that triggers a coordinated global transcriptional response in *S. pneumoniae* [8].

Bacteria accomplish essential DNA processes, such as replication and transcription, in a
way that has to be compatible with the compaction, up to 1000-fold, of their chromosomes [9].
DNA compaction results from the level of Sc and the binding of nucleoid-associated proteins
(NAPs) [10]. Sc is homeostatically maintained by the opposing activities of relaxing DNA
topoisomerases (TopIV and TopoI), and by gyrase that introduces negative supercoils. DNA
compaction, at the kilobase-size range, generates isolated loops that coil up around themselves
forming Sc-domains. Sc regulates transcription in bacteria, and, together with promoter
sequences, acts as a *cis* regulator. Regulation *in trans* is mediated by proteins, both structural
and regulatory, which target several genes, such as the NAPs [11]. Alternatively, these proteins
bind to specific promoter regions and either facilitate or inhibit the interaction of RNA poly-
merase (RNAP) [12]. In *S. pneumoniae*, the transcriptomic response to DNA relaxation by
inhibition of gyrase with novobiocin or to hypernegative Sc by inhibition of TopoI with SCN,

has revealed Sc-domains. Genes within these domains have a coordinated transcription and similar functions. Decrease of Sc mediated by novobiocin modulates the transcription of 37% of the genome. The majority (>68%) of these genes are grouped in 15 Sc-domains with either up-regulation (UP domains) or down-regulation (DOWN domains) [13]. The increase in Sc mediated by SCN modulates the transcription of 10% of the genome, with 25% of these genes grouped in 12 Sc-domains [8,14]. The position of Sc-domains in the chromosome, which are detected either by DNA relaxation (novobiocin) or by hyper-Sc (SCN), nearly overlap, supporting the organization of the chromosome into topological domains. There are other types of gene domains, not responding to Sc changes [15], such as pvNR (position-variable nonregulated), pcNR (position-conserved nonregulated) and AT-rich. AT-content is higher in UP domains than in DOWN domains. UP, DOWN, and pcNR domains are enriched in essential genes and those of the central metabolic network, while genes of AT-rich domains have the lowest transcription levels and may play a structural role [16].

Sc homeostasis in bacteria mainly occurs by regulation of transcription of the topoisomerase genes. In *Escherichia coli*, under DNA relaxation, transcription of the TopoI gene (*topA*) decreases [17], while transcription of gyrase genes (*gyrA*, and *gyrB*) increases [18–20]. In this bacterium, topoisomerase gene expression is also affected by NAPs through alteration of Sc [21, 22]. *S. pneumoniae* lacks most of the NAPs found in *E. coli*, with the exception of HU [23] and SMC [24]. In *S. pneumoniae* Sc is controlled by transcriptional regulation of their topoisomerase genes: *gyrA* and *gyrB* (gyrase), *parC* and *parE* (TopoIV), and *topA* (TopoI). The transcription levels of *topA* in homeostasis, i.e., under conditions that allow cell growth and the recovery of Sc, correlates with the induced variation in the density of Sc [8]. The regulation of *topA* transcription, located in a DOWN Sc-domain [13], is essential for the maintenance of Sc. Likewise, the transcription of *gyrB* is also regulated by their strategic chromosomal location in an UP Sc-domain [16].

It is clear that Sc controls transcription, however, transcription is, at the same time, a major contributor to the level of Sc. The twin supercoiled-domain model proposes that negative Sc and positive Sc domains are transiently generated, respectively behind and ahead of the moving RNAP [25]. *In vitro* studies have supported this model and a role for TopoI in the prevention of R-loop formation, which, otherwise, can interfere with transcription elongation [26]. In R-loops, the RNA is hybridized with its DNA template region, leaving the non-template strand unpaired [27–29]. In addition, a direct interaction between the *E. coli* TopoI with RNAP enzyme has been detected *in vitro* [30,31]. This interaction could bring TopoI relaxing activity to the site of transcription, potentially having an activating role in this process. A recent study using ChIP-Seq has showed co-localization of RNAP, TopoI and gyrase on the active transcriptional units of *M. tuberculosis* [32]. In this study we have analyzed the interaction between TopoI and RNAP of *S. pneumoniae*. We analyzed the *in vitro* physical interaction between these enzymes and the *in vivo* interaction by using ChIP-Seq, which detect the genome occupancy of TopoI and RNAP. Our results suggest a direct role of TopoI in the process of transcription carried out by RNAP.

## Results

### Direct interaction between TopoI and RNAP detected by surface plasmon resonance (SPR)

TopoI and RNAP holoenzyme of *S. pneumoniae* were purified (Fig 1A) and their potential physical interaction was analyzed by the Biacore SPR method. The RNAP core in Gram-positive bacteria consists of seven subunits, four of them are essential (α2ßß') and the rest are considered accessory subunits (δ, ε, and ω) with supportive roles in transcription [33]. The sigma factor binds to RNAP core to facilitate the recognition of the promoter region by RNAP and

**A**

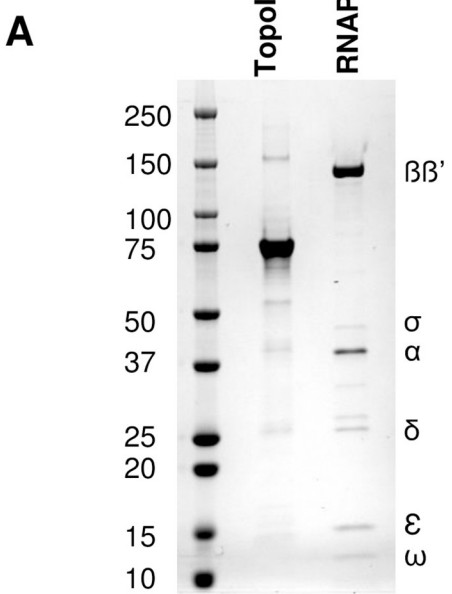

**B**

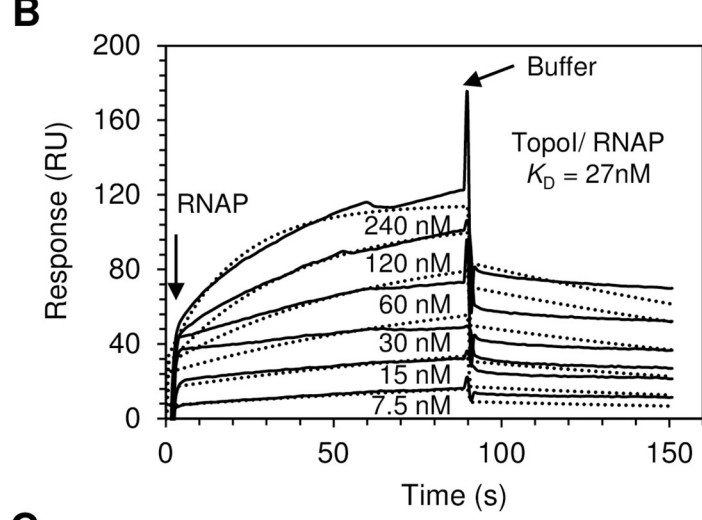

**C**

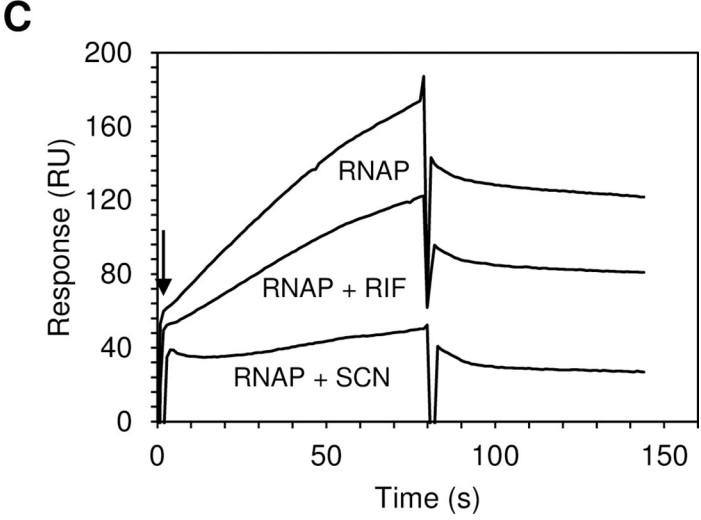

**Fig 1. Direct interaction between TopoI and RNAP detected by SPR.** (A) SDS-polyacrylamide gel electrophoresis (4–20%) of purified TopoI and RNAP used in SPR assays. Five μg of TopoI and RNAP were loaded. The gel was stained with Coomassie blue. (B) Sensograms for SPR measurements, which were corrected after subtracting the background signals (solid lines). Fitted kinetics (dotted lines) were determined using Biacore X100 Software. Concentrations of RNAP injected over TopoI are shown. (C) Sensograms for SPR measurements in the presence of inhibitors of RNP (RIF) or TopoI (SCN). Interaction of immobilized TopoI and RNP (240 nM) alone or preincubated with 1 x MIC of RIF or 1 x MIC SCN are shown.

initiate transcription. RNAP core plus the sigma factor constitutes the RNAP holoenzyme. For SPR assays, TopoI was immobilized on a sensor chip via amino coupling and RNAP holoenzyme solutions ranging from 7.5 to 240 nM were injected sequentially for measurements. Since the rate of dissociation of TopoI-RNAP complex was slow, the kinetics for this interaction was performed by regenerating the sensor. The kinetic analysis of the sensorgrams revealed that TopoI bound to RNAP with a KD value of 27 nM (Fig 1B), indicating that these two enzymes strongly interact *in vitro*. To confirm the specificity of the interaction, inhibitors of RNAP (Rif) and TopoI (SCN) were used. TopoI was immobilized on a new CM5 sensor chip and solutions of RNAP holoenzyme (240 nM) alone or incubated with 1 x MIC Rif or SCN were injected sequentially (Fig 1C). The interaction between enzymes was differentially affected by these drugs. In the absence of drugs, the response reached 172.9 RU that decreased to 122.2 RU in the presence of Rif (29. 3%). In contrast, SCN dropped the value to 40.9 RU, representing a decrease of 76.3%.

## Genome-wide association of RNAP and TopoI at promoter regions

In order to investigate the interaction between TopoI and RNAP *in vivo*, we carried out ChIP-Seq assays. To permit optimal growth conditions, we used exponentially growing cultures, which were either untreated or treated for 15 min with subinhibitory concentrations (0.5× MIC) of RNAP or TopoI inhibitors, Rif and SCN respectively. We analyzed RNAP and TopoI occupancy at promoter regions using the multiBigwigSummary tool [33], from 100 bp upstream to 100 bp downstream the start codon of each gene. By using a filter of enrichment ratio (ER) higher than 2, the promoter of 435 and 356 genes were enriched for RNAP and TopoI, respectively (Fig 2A), being 129 genes enriched for both RNAP and TopoI. To compare the expression level of the RNAP- and TopoI-enriched genes, we used our recently published data of RNA-Seq experiments in exponential cultures grown under the same conditions [8]. These data were normalized by gene size (kb) using the reads per kilobase per million mapped reads (RPKM). Taken these data as a reference, the average RPKM of the 306 RNAP-exclusive genes (15441) and the 129 common genes (17340) was higher than that of the transcriptome (7056), with *P*-values of 1.24E-13 and 8.4E-11, respectively. Genes involved in protein synthesis were more represented among the RNAP-enriched genes than in whole genome. Enrichment of DNA metabolism genes was observed for TopoI. On the other hand, genes encoding mobile genetic elements and transporters were less represented among the genes covered by RNAP or TopoI (Fig 2B).

We also observed changes in TopoI occupancy at promoter regions under SCN treatment (Fig 2C). Although the total number of TopoI-enriched gene promoters was similar, SCN caused TopoI occupancy to displace from 207 genes to 186 new gene promoters (Fig 2C). Inhibition of RNAP with Rif also affected TopoI location (Fig 2D). The total number of TopoI-enriched gene promoters diminished 3.8 times under Rif treatment (from 356 to 94), indicating that RNAP activity favours TopoI binding to promoters. As expected, treatment with Rif affected RNAP occupancy (Fig 2E), while most of the promoters remained covered by RNAP under Rif treatment, new promoters appeared. The effect of SCN on the promoter occupancy

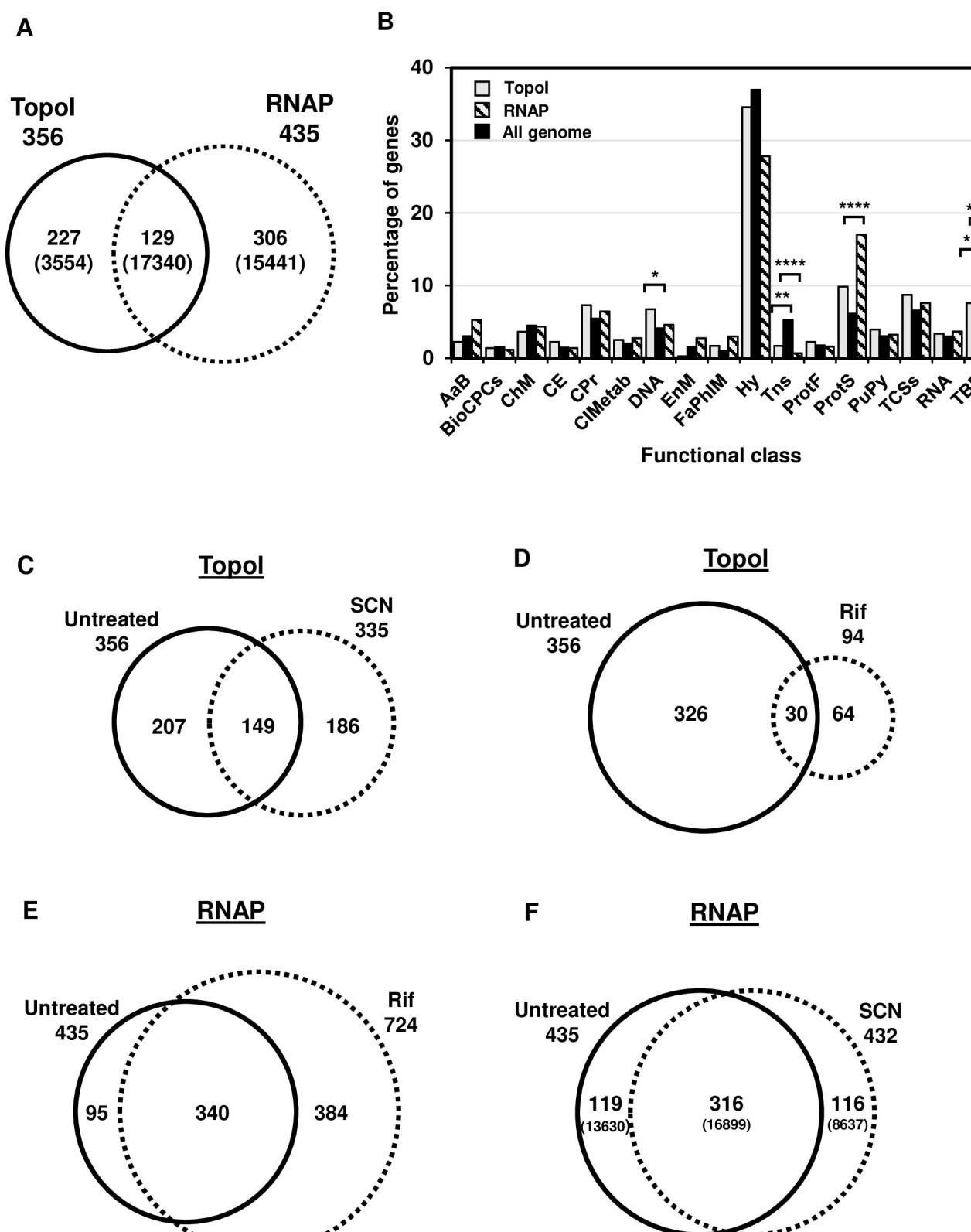

**Fig 2. TopoI and RNAP enriched genes at promoter regions.** A) Venn diagram representation indicating the number of promoters. Numbers in parenthesis indicate the average RPKM values. B) Classification of enriched genes by functional classes. Distribution of RNAP-enriched promoters

(striped boxes) and TopoI-enriched (gray columns) compared with that of the whole genome (black columns) according to the biological function of their genes. Functional classes: AaB, amino acid biosynthesis; BCPCs, biosynthesis of cofactors, prosthetic groups, and carriers; ChM, carbohydrate metabolism; CE, cell envelope B/D of murein sacculus and peptidoglycan; CPr, cellular processes; CIMet, central intermediary metabolism; DNA, DNA metabolism; EnM, energy metabolism; FaPhlM, fatty acid and phospholipid metabolism; Hy, hypothetical proteins; Tns, Mobile and extrachromosomal element functions; ProtF, protein fate; ProtS, protein synthesis; PuPy, purines, pyrimidines, nucleosides, and nucleotides; TCSs, two-component systems; RNA, RNA metabolism; TBP, transport and binding proteins. Statistical significance Chi-square test, two tailed: * $P \leq 0.05$, ** $P \leq 0.01$, *** $P \leq 0.001$ **** $P \leq 0.0001$. C) Comparison of TopoI-enriched genes in cultures untreated or treated with SCN. D) TopoI-enriched genes untreated or treated with Rif. E) RNAP-enriched genes untreated or treated with Rif. F) RNAP-enriched genes untreated or treated with SCN.

by RNAP was lower than that of Rif (compare Fig 2E and 2F). Moreover, SCN treatment did not change the proportional representation of gene functional classes.

## Uneven location of RNAP and TopoI along genes

ComputeMatrix followed by plotProfile tools [33] was used to scan the occupancy of RNAP and TopoI throughout genes (Fig 3). Regions analysed covered from 100 bp upstream the ATG start codon to the stop codon, and the gene size was scaled to 800 bp. Considering all genes, RNAP occupancy showed a peak upstream the start codon, and a regular distribution along the open reading frame (Fig 3A). As expected, occupancy of RNAP in highly expressed genes (HE, the third most expressed, RPKMs > 3579) was clearly higher than in lowly expressed genes (LE, the third least expressed, RPKMs < 843). When cultures were treated with Rif, higher peaks centered around positions 50 bp (Rif at 0.5× MIC) or 0 bp (Rif at 1× MIC) were observed (Fig 3C), suggesting RNAP became trapped at the promoter upon inhibition. However, some elongation was allowed at subinhibitory concentrations of Rif (0.5× MIC). It is known that Rif binds the RpoB subunit of RNAP in both the close and open RNAP-DNA complexes, but does not bind the elongation complex, blocking extension of the RNA from 2–3 nt at high concentration (>6× MIC) [34]. However, the low Rif concentration used in our experiments, allowed some RNA elongation (Fig 3C). This effect has been also observed previously in *E. coli* using antibodies against RpoB [35]. Likewise, in HE genes, the positioning profile of RNAP was similar while the occupancy was higher (Fig 3G). SCN is a specific inhibitor of the TopoI cleavage reaction [5,7]. Inhibition of TopoI by SCN prevented the occupancy of RNAP around the ATG codon, and caused a moderate increase of the RNAP occupancy at the coding region (Fig 3E and 3G).

As RNAP, TopoI was also preferentially positioned upstream the ATG codon in untreated cultures (Fig 3B). However, its occupancy along the gene body was lower than that of RNAP and a clear effect of the level of gene expression was no observed. As TopoI relieves negative Sc generated during transcription, it could be expected that as the distance between ATG and transcription termination site increases in long genes, a higher presence of TopoI at positions further away from the ATG would be detected. We observed that TopoI occupancy was high around the end of long genes (the third longer genes, >960 bp) compared to all genes or short genes (the third shorter genes, <522 bp) (S2 Fig).

Inhibition of TopoI by SCN leaded to a sharp peak near the ATG codon that was higher in HE genes (Fig 3D and 3H), suggesting immobilization of the enzyme in a way dependent on transcription efficiency. In agreement with this suggestion, the inhibition of RNAP by Rif eliminated TopoI occupancy (Fig 3F and 3H).

## Location of RNAP and TopoI along genes depends on their orientation relative to replication

The orientation of protein-coding genes in *S. pneumoniae* in relation to the direction of replication is shown in Fig 4A. Gene transcription shows a preferential co-directional orientation

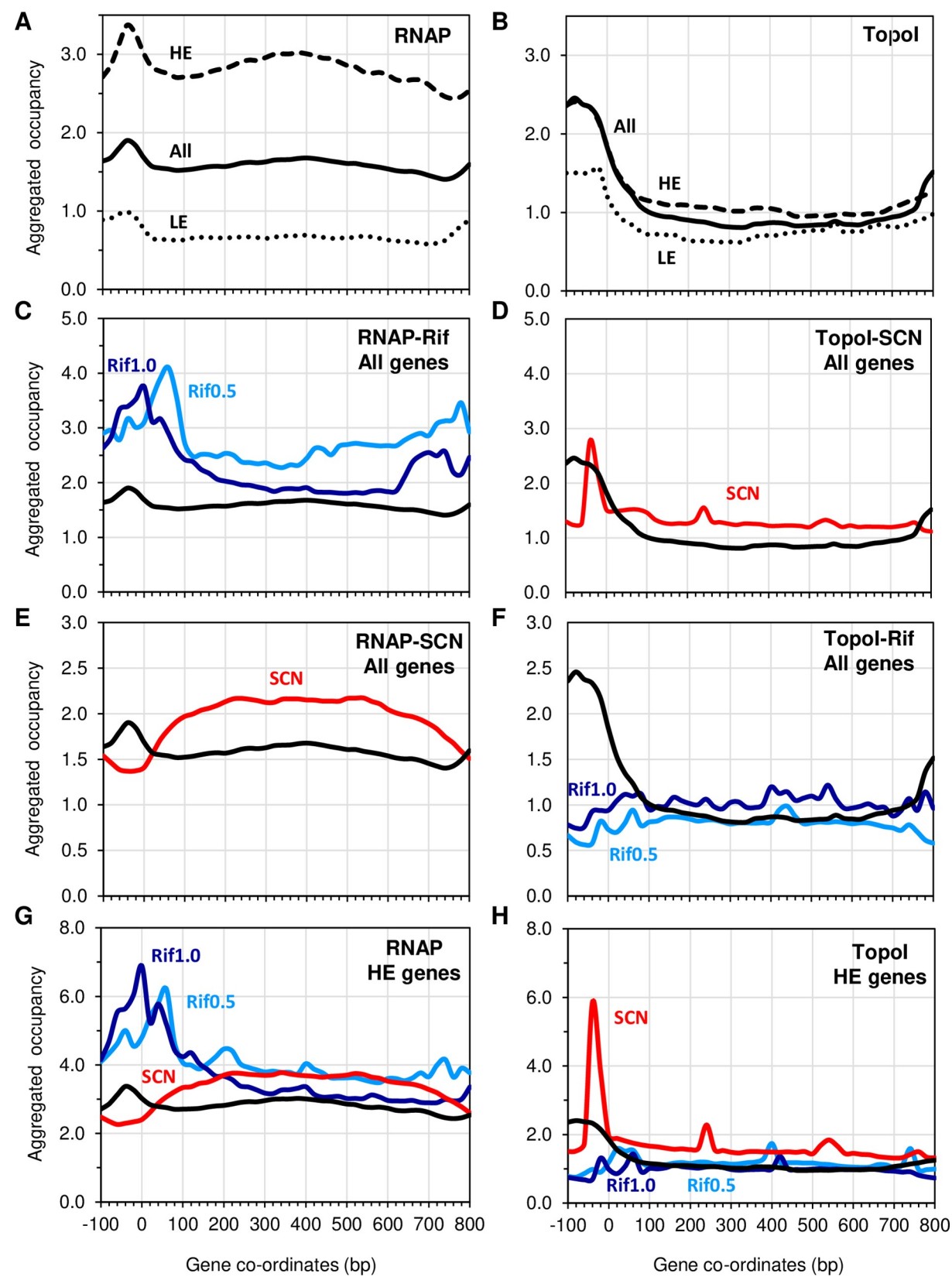

**Fig 3. Distribution profiles of RNAP and TopoI across protein-coding genes of *S. pneumoniae* in untreated and Rif- or SCN-treated cultures.** Aggregated occupancy was obtained using ComputeMatrix followed by plotProfile tools. Gene size was scaled to 800 bp and the 100 bp upstream of the ATG start codon were included in the analysis. A, B) Distribution across all, highly expressed (HE) and lowly expressed (LE) genes. C, D) Distribution of RNAP across all genes. E, F) Distribution across HE genes.

with replication. Genes transcribed from right to left (- genes) are mostly located at the right replichore, where replication proceeds in the same direction, whereas genes transcribed from left to right (+ genes) are mostly located in the left replichore. Therefore, most protein-coding genes (79.1%, 1616 out of 2043) are co-directionally oriented relative to replication in *S. pneumoniae* indicating that an evolutionary pressure prevents head-on collision between transcription and replication. In support of this, all four ribosomal RNAs (rRNA) operons present in the *S. pneumoniae* genome and 94.8% (55 out of 58) of the tRNA genes are transcribed in the same direction of replication.

The occupancy of RNAP was higher in co-directional genes than in head-on genes (Fig 4B), while the opposite was observed for TopoI (Fig 4C). These preferential occupancies were more pronounced in HE genes, for both RNAP and TopoI (Fig 4D–4G). It has been reported that TopoI is preferentially located in the intergenic regions of adjacent genes that are divergently transcribed [36,37], where it would release the negative Sc accumulated behind the transcription machinery. Our results shown in Fig 5A support this idea. TopoI was more abundant in the intergenic region of divergent (D) genes than that of convergent genes (C). We analyzed independently divergent gene pairs located in the left (DL) and in the right (DR) replichore of the genome. In DL gene pairs the left gene is the head-on oriented relative to replication, while in DR pairs the right gene is the one that is head-on oriented (black arrows). We observed differences in the distribution and amount of TopoI in the intergenic regions of these two types of gene pairs. TopoI occupancy between DL gene pairs showed a sharp peak towards the left end of the intergenic region, near the 5' end of the head-on gene. In the case of DR gene pairs, TopoI showed a broader peak towards the right end of the intergenic region, closer to the 5' end of head-on genes. Then, the preferential binding of TopoI to the intergenic region of divergent genes seems to be mediated by its binding to the promoter region of the head-on gene of the pair. We have also found that the amount of TopoI is much higher in the intergenic regions of DR than DL pairs (Fig 5A). This is likely due to the higher expression level of the head-on gene in the DR that in the DL gene pairs (Fig 5B). This would recruit more TopoI to favor transcription and release torsional stress.

## Identification of TopoI and RNAP peaks and identification of binding motifs

We used MACS2 peak calling tool to identify 154 peaks for TopoI and 341 for RNAP (-$\log_{10} P$ value >30). The peaks corresponding to TopoI mapped at 131 genes and those corresponding to RNAP mapped at 291 genes, among them, 52 were common (Fig 6A). Genes containing either RNAP peaks or both RNAP and TopoI peaks showed average RPKM values of 11992 and 13535, respectively. These values are higher than the average RPKM for the genome (7056), with *P* values of 1.7E-5 and 5.8E-3, respectively. Regarding function, genes involved in protein synthesis were highly enriched among the genes that contained peaks for only RNAP (13.0% of the genes versus 6.1% genome-wide). Since the genes involved in protein synthesis are highly expressed, these results suggest a preferential binding of RNAP to highly expressed genes. Summits of RNAP and TopoI peaks were mapped with respect to the initiation codon of genes (Fig 6B). A sharp distribution centered at the ATG codon was observed for TopoI, while position of RNAP peak summits showed a broader distribution, which was more

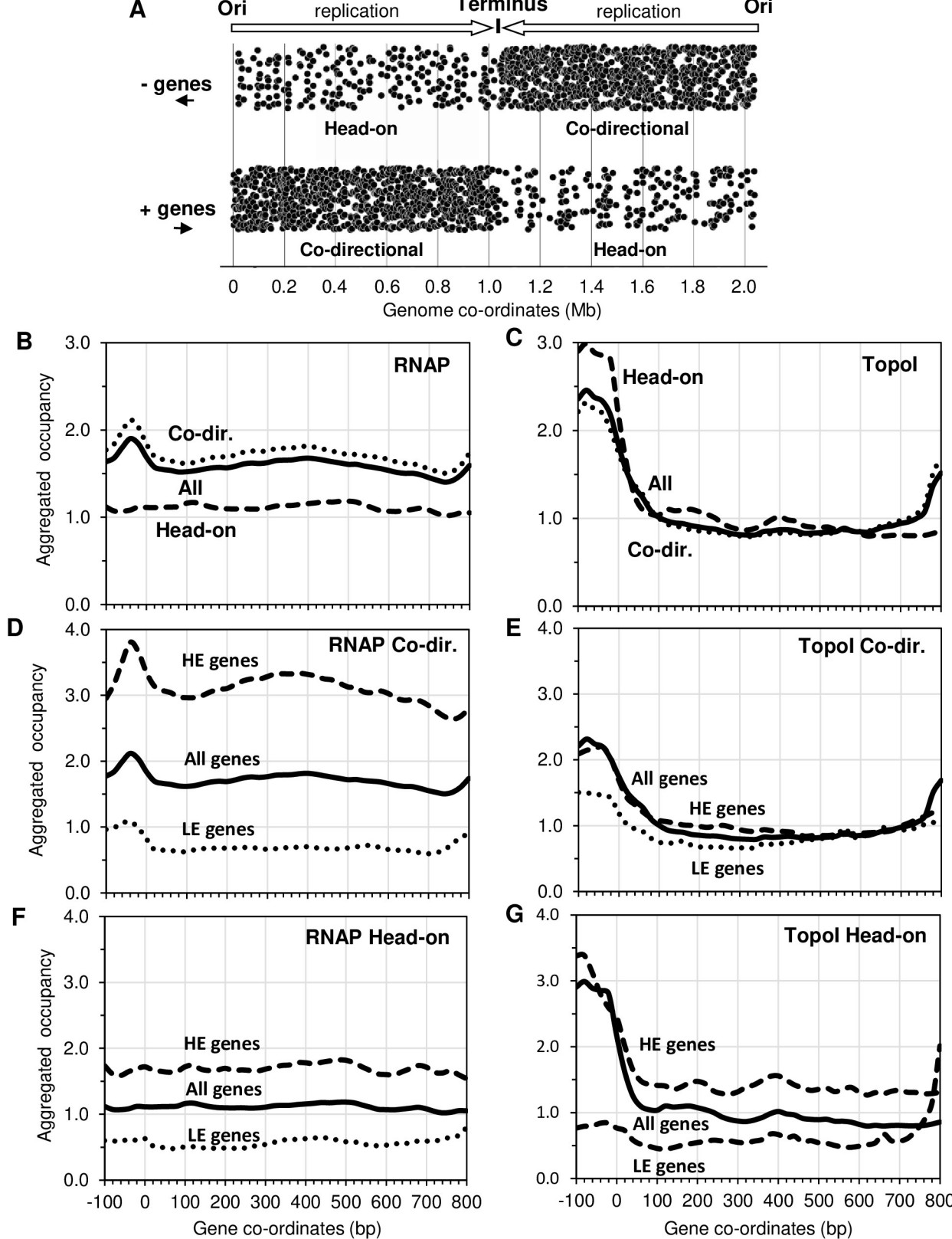

**Fig 4. Distribution profiles of RNAP and TopoI along protein-coding genes considering transcription orientation and replication.** A) Location of genes in the genome depending of its transcription orientation in relation with replication from the origin (Ori) to the terminus.

Genes transcribed from right to left (- genes) are indicated in the upper part of the graph, those transcribed from left to right (+ genes) are indicated in the lower part of the graph. Co-directional genes transcribe in the same direction as replication. Head-on genes transcribe in the opposite direction than replication. B, C, D, E, G) Aggregated occupancy in co-directional and head-on genes obtained using ComputeMatrix and plotProfile tools. Gene size was scaled to 800 bp and the 100 bp upstream of the ATG start codon were included in the analysis. All, highly expressed (HE) and lowly expressed (LE) genes were considered.

displaced towards positions downstream the ATG. These results suggest that TopoI is located upstream of RNAP, probably removing the negative Sc generated in the transcription process.

The MEME Suite was used to recognize consensus motif sequences in the peaks. Two motifs present in a high number of peaks were identified for TopoI (Fig 6C). The AT-rich motif 1 was the most abundant, being present in 44.8% of the peaks. The median of the locations of all sites corresponding to motif 1 was +4 bp relative to the ATG codon. The A-rich sequence of motif 1 was preferentially located in the coding strand of the genes (82 out of 102 sites, black dots in Fig 6C). This motif was present in 50 intergenic regions: 6 associated to promoter regions; 20 to ribosome-binding sites; 1 to the ATG codon.

Motif 2 was identified in 18.2% of the peaks and contained the previously reported -10 extended sequence (5´-TnTGnTATAAT-3´), present in several pneumococcal promoters [38]. The median location of motif 2 sites was -21 bp relative to the ATG start codon. Most motif 2 sites (23 out of 32) were located between position -100 bp and the ATG codon and its orientation was compatible with a role as -10 extended promoter. We analyzed how common motif 2 was at promoters not enriched by TopoI (ER<2) compared to TopoI-enriched promoters (ER>2). We used FIMO program to search for significant occurrences ($P$-value<1E-6) of this motif in each subset of promoters. Motif 2 was significantly more frequent ($P$-value <0.0001) at TopoI-enriched promoters (15.7%, 56/356) than at the promoters not enriched by TopoI (5.1%, 87/1687).

Regarding RNAP, we found two motifs present in a high number of peaks (Fig 7A). A total of 97 peaks (28.4%) contained motif 1 sites, and 144 peaks (44.2%) contained motif 2. These

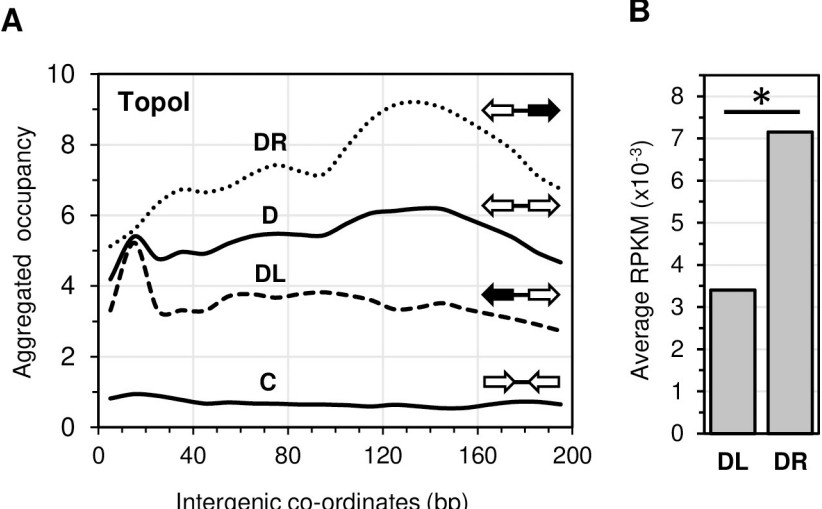

**Fig 5. Differential TopoI occupancy along the intergenic regions between contiguous protein-coding genes and its relation to the expression level of the head-on gene.** A) Aggregated occupancy of TopoI in the intergenic regions of divergent genes (D), convergent genes (C), divergent gene pairs located in the left replichore (DL) and divergent gene pairs located in the right replichore (DR) as indicated in Fig 4A. Head-on genes are indicated with black arrows in DL and DR. B) Average RPKM of the head-on gene of DL and DR gene pairs. Statistical significance: *$P$ < 0.05.

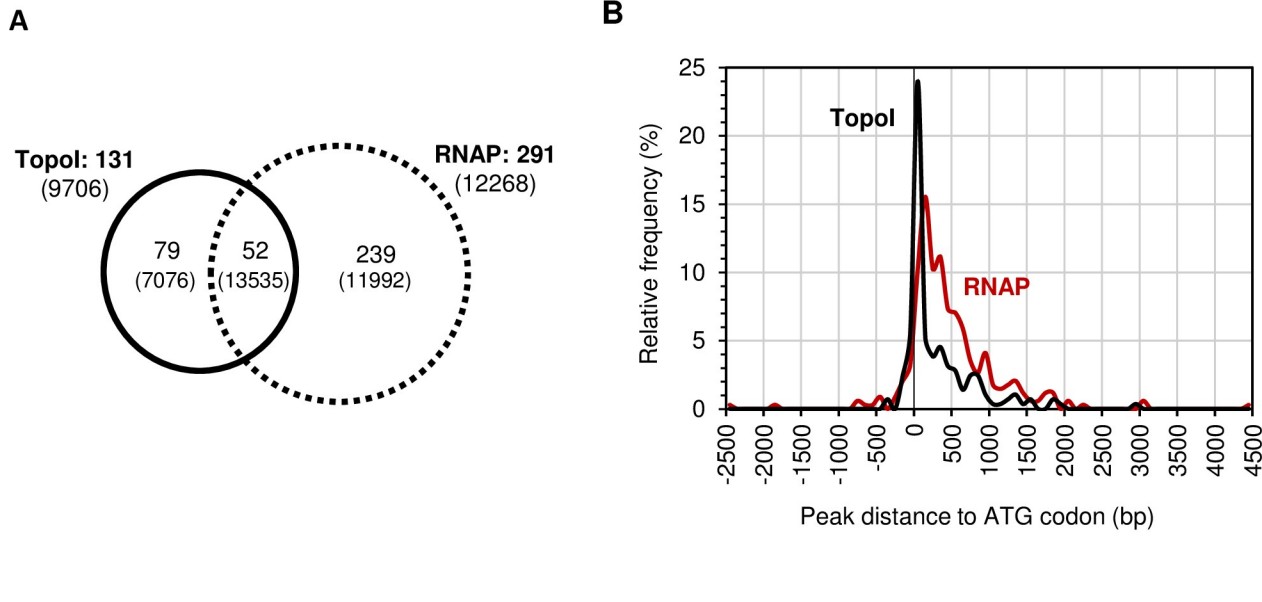

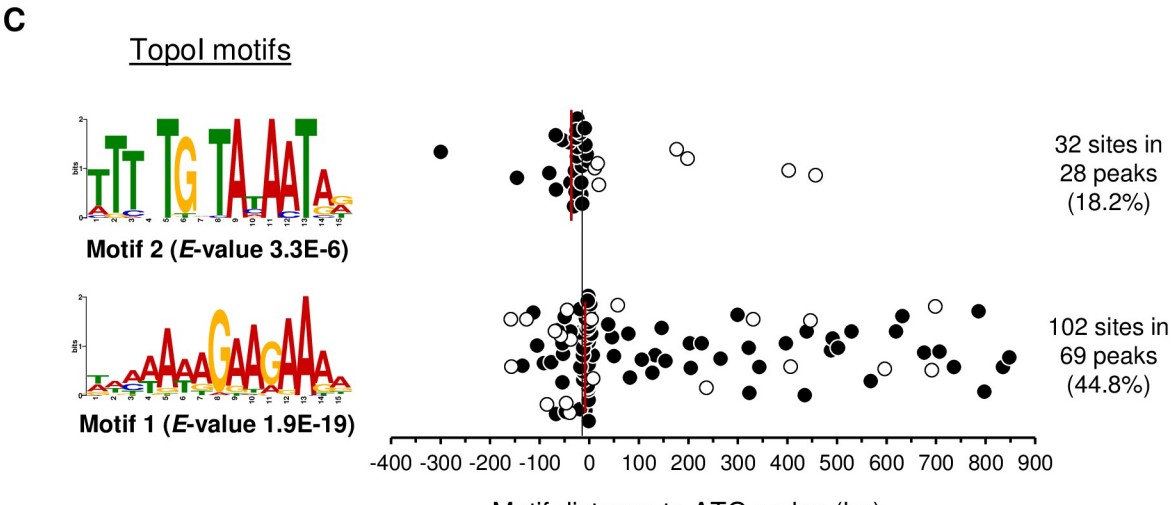

**Fig 6. TopoI and RNAP peak detection and identification of binding motifs.** Genes where RNAP and TopoI peaks were detected using MACS2. Only those peaks with *P* values <1E-30 were considered. A) Venn diagrams showing the number of genes, and, in parentheses, the average RPKMs. B) Position of the summits of TopoI and RNAP peaks. **C**) Position of the indicated TopoI binding motifs identified using MEME suite [51]. In motif 2, black dots correspond to intergenic regions and white dots to intragenic regions. In motif 1, black dots correspond to motifs located in the coding strand, while white ones correspond to motifs located in the non-coding strand. Four positions are not represented since their location was >900 bp of the gene. Median. The black line indicates the ATG start codon and the red lines the medians of positions of each motif.

motifs were mostly located in intragenic regions of highly expressed genes. When the same analysis was performed in cultures treated with Rif at 1× MIC, two different motifs were detected (Fig 7B). Motif 1, present in 85 peaks (42.5%), was mainly located in intragenic regions. On the other hand, Motif 2 corresponded to the -10 extended promoter sequence and, as expected, was mainly located (82.2%) at intergenic regions. Comparison of this motif to motif 2 of TopoI using Tomtom tool indicates they are equivalent motifs (E-value = 2.3E-7, Fig 7C)

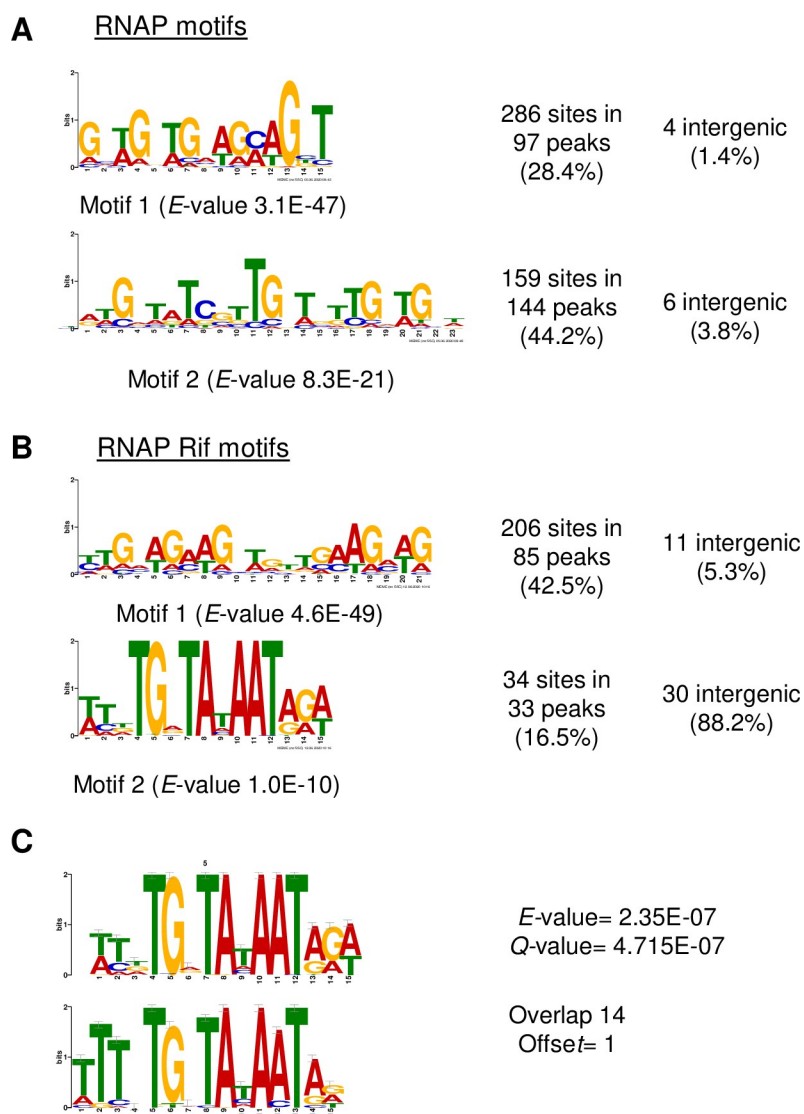

**Fig 7. Identification of RNAP binding motifs using MEME.** Motifs discovered in untreated A) or Rif-treated cultures B). C) Comparison of motifs 2 of RNAP and TopoI using Tomtom tool [52].

## Discussion

To understand the interplay between transcription and Sc regulation by TopoI *in vivo*, we studied the genome-wide association of RNAP and TopoI in the Gram-positive pathogenic bacteria *S. pneumoniae*, which has a single type-I topoisomerase and a reduced number of NAPs. We have previously shown that the homeostasis of Sc in *S. pneumoniae* depends mainly on the transcriptional control of their topoisomerases, especially TopoI [8,13]. Our ChiP experiments showed that RNAP is preferentially located at promoter regions (Fig 2A) of highly expressed genes, and promoters bound by both TopoI and RNAP had higher levels of expression than those bound by only RNAP (Fig 2A). These results might indicate that TopoI plays a role in the activation of transcription. Furthermore, mapping of summit peaks confirms that TopoI is located close to and upstream of RNAP (Fig 6B), supporting a role for TopoI in eliminating the negative supercoils generated upstream the transcription complex. A similar

conclusion has been recently reached in *M. tuberculosis* [32]. This is further supported by the MEME motifs found. A TopoI binding motif that corresponds to the -10 extended sequence (5´-TnTGnTATAAT-3´), commonly found at pneumococcal promoters [38], is present in 28 TopoI peaks (Fig 6C). This same motif was found for RNAP when it was blocked at the promoter by Rif (Fig 7B and 7C). Likewise, scanning the occupancy through genes showed a close proximity between TopoI and RNAP (Fig 3A and 3B). This proximity could be mediated by a physical interaction between TopoI and RNAP, which is supported by the observed tight *in vitro* interaction between both proteins (Fig 1B). RNAP-TopoI *in vitro* physical interaction has also been reported for its homologs in *E. coli* [30,31]. We observed a tighter association in *S. pneumoniae* ($K_D$ = 27 nM) than that observed in *E. coli* ($K_D$ = 93 nM), and a slower dissociation, probably because we used the RNAP holoenzyme, while only the core enzyme was used in *E. coli*. In addition, structure-based modelling of RNAP-TopoI interaction in *E. coli* has identified the residues of TopoI and subunit RNAP β' [31] involved in the interaction. However, none of these residues are conserved in the pneumococcal proteins. Likewise, it has been reported that the residues utilized by *M. tuberculosis* TopoI to interact with RNAP are different from those used by *E. coli* TopoI [31,39]. This shows that different bacterial species utilize distinct mechanisms for TopoI-RNAP interaction. Rif, which inhibits the pneumococcal RNAP β subunit [40], prevented TopoI-RNAP interaction (Fig 1C), suggesting a role for RNAP β subunit in such interaction. More structural studies in the pneumococcus are needed to characterize this interaction.

The cross effect of inhibition of RNAP on TopoI positioning and *vice versa*, gave us additional information about the interplay of RNAP and TopoI. Inhibition of RNAP with Rif caused TopoI to no longer occupy the position upstream the ATG codon (Fig 3F). There was also a cross effect of inhibition of TopoI with SCN on RNAP occupancy (Fig 3E). Interestingly, a role for TopoI in RNAP positioning across genes was also evident as TopoI inhibition with SCN displaced the preferential location of RNAP near the ATG codon to positions throughout the coding region (Fig 3E). In agreement with these results, RNAP-TopoI interaction was inhibited *in vitro* by both the presence of Rif and SCN (Fig 1D). This suggests that TopoI plays a role in the formation or stability of the RNAP-DNA complex at the promoter, likely by establishing the appropriate local level of negative Sc. In addition, TopoI facilitates transcription elongation by removing negative Sc produced behind the transcription machinery. In agreement, a recent report has described an enhanced occupancy of RNAP at the coding regions when TopoI is absent [41]. Formation of R-looping induced by accumulated negative Sc, which is known to be prevented by TopoI in *E. coli* [42], might be a cause of RNAP elongation hampering after TopoI inhibition. We observed that TopoI is more enriched at gene promoters and at the end of long genes than in the gene body (Figs 3B and S2). This might be related to the higher presence of R-loops at these positions, as observed in eukaryotes [43].

When the function of genes whose promoters are covered by RNAP and TopoI is considered, those involved in protein synthesis were overrepresented among RNAP-enriched genes (Fig 2B). These included most of the genes involved in ribosomal-proteins synthesis (50 out of 62) and most of the translation factors (9 out of 11). These results are compatible with the fact that ribosomal proteins are the most highly expressed genes in bacteria and have the highest codon-adaptation indexes [44]. TopoI was also preferentially recruited to the promoter (Fig 2B) of genes involved in DNA metabolism, supporting a role of Sc in the transcriptional regulation of those genes. In fact, in *S. pneumoniae*, these genes tend to be located in Sc domains responding to relaxation with novobiocin [13]. Low representation of genes coding mobile genetic elements and transporters was observed among the genes covered by both RNAP and TopoI, suggesting they are lowly expressed, in agreement with their low codon-adaptation indexes [45].

In bacteria, head-on collisions between RNAP and the replication complex are avoided by a genome-wide bias towards co-directional orientation of genes relative to the direction of replication [46], so that promoters of most genes are proximal to the replication origin and gene termini are proximal to the replication terminus. Several studies indicate that the main selective pressure responsible for this gene organization is the preservation of genome integrity, which is at risk when transcription and replication head-on collide [47]. In *S. pneumoniae*, we also observed a strong transcription-replication co-orientation bias: near 80% of the protein-coding genes, the four rRNA operons and about 95% of the tRNA genes are co-oriented with DNA replication (Fig 4A). We have found that RNAP occupancy is higher in co-directional genes than in head-on genes, while the opposite is observed for TopoI (Fig 4B and 4C), an effect more pronounced in HE genes compared to LE genes (Fig 4D–4G). This indicates that TopoI plays a role in head-on genes, likely favouring their transcription that is compromised due to the accumulation of positive Sc when transcriptional and replication forks approach in opposite directions. Accordingly, co-directional genes are enriched in genes of pcNR domains, i.e., highly expressed genes not subjected to Sc transcriptional regulation [15] compared to head-on genes (291 of 1636 versus 47 of 409, *P* = 0.0018). The opposite is observed for genes of DOWN and UP domains, i.e., genes whose transcription is regulated by Sc (179 of 409 versus 584 of 1636, *P* = 0.0029). Since there are not significant differences in the levels of transcription of co-directional genes (average RPKM = 7300) and of head-on genes (average RPKM = 6150), these results support the role of TopoI in the transcription of head-on genes.

In agreement with previous reports [32,37], we have observed that TopoI is preferentially located in the intergenic regions of divergent gene pairs compared to those of convergent gene pairs (Fig 5A). TopoI would be recruited to these regions to release the negative Sc accumulated behind divergent RNAPs, while positive Sc generated ahead convergent RNAPs would be released by the DNA gyrase [32,41]. In agreement with the preferential TopoI location at the promoters of head-on oriented genes (Fig 4C), distribution of TopoI across the intergenic regions of divergent genes is not regular, but it is preferentially located near the promoter of the gene that is oriented head-on relative to replication (Fig 5A). Therefore, binding of TopoI to the promoters of head-on genes contributes to its preferential location in intergenic regions of divergent genes pairs. In addition, the amount of TopoI was higher at the intergenic regions of DR than DL gene pairs (Fig 5A). This is likely due to the higher expression of the head-on gene in the DR pairs (Fig 5B) than in the DL pairs, which would recruit more TopoI to release negative Sc and favor transcription.

In conclusion, genome-wide proximity between RNAP and TopoI occurs in *S. pneumoniae* mainly at gene promoter regions. This stimulates transcription initiation and, likely, elongation by regulating the level of negative Sc upstream the transcription complex.

## Material and methods

### Microbiological methods and genetic constructions

Cultures of *S. pneumoniae* R6 strain were grown at 37˚C in a casein hydrolysate-based liquid medium (AGCH) containing 0.2% yeast extract and 0.3% sucrose [48]. R6 MICs (minimal inhibitory concentrations) are of 0.015 μg/ml for Rif (0.02 μM), of 5.2 μg/ml (16 μM) for SCN. A strain containing the ß' subunit of pneumococcal RNP with a C-terminal 10-His-tag was constructed, based on a strategy previously described [49]. A 530-bp fragment containing the 3′end of *rpoC* with a 10-His-tag was amplified by PCR using DNA of strain R6 and Pfu DNA polymerase (Biotools) using primers DAM254 and DAM211 [49], the latter containing an EcoRI restriction site. The PCR product was digested with EcoRI and cloned into the EcoRI/SmaI sites of plasmid pJDC9, which carries an erythromycin-resistance determinant and

replicates in *E. coli* but not in *S. pneumoniae*. Strain R6 was transformed with the recombinant plasmid carrying *rpoC*-10-His-tag (pJDCrpoC) and selection was made with 2 μg/ ml of erythromycin. This allowed the plasmid to be integrated at *rpoC* by insertion duplication. The resulting strain was named R6RpoCHis.

## Purification of proteins

Purification of the His-tagged pneumococcal RNAP was carried out from 4 liters of a culture of R6RpoCHis as previously described [49]. Briefly, R6RpoCHis was grown in until $OD_{620nm}$ = 0.1, cells were harvested by centrifugation at 4000 x g for 15 minutes and pellet was washed with rinse buffer (10 mM TrisHCl pH 8.0, 300 mM NaCl, 20% glycerol, 10 mM $MgCl_2$, 5 mM ß-mercaptoethanol) once and stored at -80˚C. The pellet was thawed and resuspended in 10 ml of lysis buffer (rinse buffer containing 1 mM phenylmethylsulfonyl fluoride and 0.1% Triton X-100). Five μg of DNase I were added and the mix was incubated 10 min at 37˚C. The lysate was centrifuged at 4000 x g for 40 min and the supernatant was applied to a 0.25 ml Ni-nitriloacetic acid resin column previously equilibrated with lysis buffer. Column was washed 3 times with 7 ml of lysis buffer containing 5 mM imidazole and 3 times with 5 ml of lysis buffer containing 45 mM imidazole. Proteins were eluted with two sequential steps: 1 ml of lysis buffer containing 105 mM imidazole; 1 ml of lysis buffer containing 205 mM imidazole. The two elution fractions were pooled, concentrated with a 3K Amicon Ultra Centrifugal Filter (Millipore) and dialyzed 3-fold against 400 ml of binding assay buffer (20 mM Tris HCl pH 7.5, 100 mM NaCl, 10 mM $MgCl_2$, 0.1 mM EDTA, 1 mM dithiothreitol, 0.005% Tween-20 and 5% glycerol).

TopoI was purified from a culture of *E. coli* M15 (pREP4)/pQE-SPNtopA strain in which TopoI is His-tagged at the N-terminus. The protein was purified by affinity chromatography in a Ni-NTA (Quiagen) column following manufacturer's instructions as described previously [5].

## Surface plasmon resonance (SPR)

Biacore X100 (GE Healthcare) device was used to determine binding of TopoI and RNAP holoenzyme of *S. pneumoniae*. A CM5 sensor chip (GE Healthcare) was used to immobilize about 3000 RU of TopoI as ligand according to the manufacturer protocol. Immobilization scouting was performed with pH 5.0 acetate. Concentrations of RNAP holoenzyme ranging from 7.5 nM to 240 nM were flowed at a rate of 50 μl/ min over the TopoI-immobilized surface of the sensor chip. Surface plasmon resonance (SPR) assays were performed at 25˚C in binding assay buffer (20 mM Tris HCl pH 7.5, 100 mM NaCl, 10 mM MgCl2, 0.1 mM EDTA, 1 mM dithiothreitol, 0.005% Tween-20 and 5% glycerol). Regeneration of the sensor surface was carried out with 1M NaCl with a flow rate of 10 μl/ min and a contact time of 60 seconds. Two independent assays were performed to calculate the KD value for association of pneumococcal TopoI and RNAP. Binding kinetics were determined using Biacore X100 Software.

## Purification of antibodies for Chromatin immunoprecipitation (ChIP-Seq) assay

Polyclonal rabbit antibodies against TopoI were obtained as previously described [8]. To obtain antibodies against RpoB, the protein with an N-terminal 6-His tag was overproduced from strain *E. coli* XL1 (pQERPOB, carrying 6His-*rpoB*) and purified by Ni-affinity chromatography. Polyclonal rabbit antibodies were obtained from Davids Biotechnologie from 0.5 mg of protein extracted from SDS-gel following a 28-day SuperFast immunization protocol. Subsequent purification of TopoI and RpoB antisera was performed in three stages. First, they

were enriched on IgGs by binding of the antisera to HiTrap Protein G HP column (GE Health-care). The second step was removing of non-specific IgGs using a BSA-precoupled glyoxal aga-rose beads column. The third step was the antigenic specific affinity purification with H6TopA or H6RpoB precoupled glyoxal agarose beads columns. Purified antibodies were dialyzed in PBS 1X buffer and concentrations determined by Bradford assay obtaining 0.12 mg/ ml and 0.19 mg/ ml of anti-RpoB and anti-TopA antibodies, respectively. To check purity a 4–20% polyacrylamide gel, was loaded, electrophoresed, and stained with Coomassie blue (S1A Fig). Titration was performed by ELISA (S1B Fig).

## ChIP-Seq assay

Cultures of 90 ml of strain R6 untreated or treated with rifampicin (Rif) or SCN grown to $OD_{620nm} = 0.4$–$0.6$ ($6.5$–$9.7 \times 10^9$ cells) were mixed by inversion with 9 ml of fixing solution (50 mM Tris [pH 8.0], 100 mM NaCl, 0.5 mM EGTA, 1 mM EDTA, 11% [weigthl/ vol] formal-dehyde) and incubated at room temperature for 30 min to give a final formaldehyde concen-tration of 1%. To stop crosslinking, 9 ml of quenching solution (1.25 M glycine, 50 mM Tris [pH 8.0], 100 mM NaCl, 0.5 mM EGTA, 1 mM EDTA) was added and the mix was incubated for 30 min at 4˚C. Cells were then washed with phosphate-buffered saline three times to remove formaldehyde. Cellular lysis and DNA shearing were performed by applying 12 cycles (30 sec ON/ 30 sec OFF) of ultrasound with Bioruptor (Diagenode) in 0.1 ml tubes. The sheared DNA sized from 200 to 700 bp. A whole cell extract control fraction containing $5 \times 10^8$ cells was kept. Fragmented DNA from the remaining cells ($3 \times 10^9$) was immunopre-cipitated with either 6 μg (anti-TopA) or 3 μg (anti-RpoB) of purified antibodies precoupled to 100 μl protein G Dynabeads (Invitrogen).

## ChIP-Seq library construction and sequencing

For library construction the TruSeq ChIP Sample Prep Kit from Illumina was used following manufacturer's instructions. Basically, DNA fragments of 150–250 bp were selected after repairing ends, adenylating 3' ends, and ligating indexed paired-end adapters. The selected DNA fragments were enriched by PCR with the PCR primer cocktail that anneals to the ends of the adapters. Quality control of the library was determined with a 2100 Bioanalyzer Instru-ment (Agilent Technologies). Sequencing was accomplished with a NextSeq 500 platform using a $1 \times 75$ base run.

## ChIP-Seq data analysis

Analysis of ChIP-Seq data was carried out using the web-based platform Galaxy [50]. Quality of raw sequence data was analysed with FASQC tool. Reads were mapped against the *S. pneu-moniae* R6 genome (ASM704v1) using BWA software package in simple analysis mode. The ratio between the BAM files from each ChIP-Seq experiment and its corresponding mock was obtained using bamCompare tool (bin size of 10 bases). The output files in BigWig format were used to calculate the average scores for specific genomic regions (genes and gene promot-ers) using multiBigwigSummary tool [33]. Occupancy of RNAP and TopoI throughout pro-tein-coding genes and intergenic regions was carried out using ComputeMatrix followed by plotProfile tool. The BAM files from BWA read mapping from each experiment and its corre-sponding mock were used for peak calling with MACS2 tool. The MEME Suite was used to discover consensus motif sequences in the peaks.

## Supporting information

**S1 Fig. A) Purified antibodies against RpoB and TopoI.** A 4–20% SDS-PAGE loaded with 5 μg of purified antibodies and stained with Coomassie Blue. Molecular weights are indicated in kDa. H (Heavy Chain), L (Light chain). **B)** Titration ELISA of purified antibodies. 0.1 μg/well of RpoB or TopoI were adsorbed to microtiter plates and serial dilutions (1/3) of sera were tested.
(TIF)

**S2 Fig. Aggregated occupancy of TopoI along *S. pneumoniae* genes.** The procedure was as described in Fig 3. The results obtained for all genes, long genes (the third longer genes, >960 bp) and short genes (the third shorter genes, <522 bp) are shown.
(TIF)

**S1 Data. Underlying numerical data for graphs.**
(XLSX)

## Acknowledgments

We acknowledge the expert advice of Mercedes Dominguez (CNM, ISCIII, Spain) in antibody purification and Vicente Mas (CNM, ISCIII, Spain) for his advice in SPR methodology. We also acknowledge Miguel Hernández-González (Crick Institute, UK) and Pedro A. Lazo (IBMCC, CSIC) for correcting the English version and for critical reading of the manuscript.

## Author Contributions

**Conceptualization:** María-José Ferrándiz, Pablo Hernández, Adela G. de la Campa.

**Data curation:** Pablo Hernández.

**Formal analysis:** Pablo Hernández, Adela G. de la Campa.

**Funding acquisition:** Adela G. de la Campa.

**Investigation:** María-José Ferrándiz.

**Methodology:** María-José Ferrándiz, Pablo Hernández, Adela G. de la Campa.

**Project administration:** Adela G. de la Campa.

**Resources:** María-José Ferrándiz, Pablo Hernández, Adela G. de la Campa.

**Software:** Pablo Hernández.

**Validation:** María-José Ferrándiz, Adela G. de la Campa.

**Visualization:** Pablo Hernández, Adela G. de la Campa.

**Writing – original draft:** Pablo Hernández, Adela G. de la Campa.

**Writing – review & editing:** María-José Ferrándiz, Pablo Hernández, Adela G. de la Campa.

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
