## [Decision Letter · Decision Letter 0]

18 Jan 2021

Dear Adela,

Thank you very much for submitting your Research Article entitled 'Genome-wide proximity between RNA polymerase and DNA topoisomerase I supports transcription in Streptococcus pneumoniae' to PLOS Genetics.

The manuscript was fully evaluated at the editorial level and by independent peer reviewers. The reviewers appreciated the attention to an important problem, but raised some substantial concerns about the current manuscript. Based on the reviews, we will not be able to accept this version of the manuscript, but we would be willing to review a much-revised version. We cannot, of course, promise publication at that time.

If you decide to revise the manuscript for further consideration at PLOS Genetics, please aim to resubmit within the next 60 days, unless it will take extra time to address the concerns of the reviewers, in which case we would appreciate an expected resubmission date by email to plosgenetics@plos.org.

[LINK]

Please do not hesitate to contact us if you have any concerns or questions.

Yours sincerely,

Josep Casadesús

Section Editor: Prokaryotic Genetics

PLOS Genetics

Reviewer's Responses to Questions

**Comments to the Authors:**

Reviewer #1: The results in this manuscript demonstrate that high level of transcription in Streptococcus pneumoniae may require the interaction between RNAP and TopoI. As reported earlier for E. coli and M. tuberculosis, TopoI from S. pneumoniae can bind directly to RNAP. The presence of a binding motif corresponding to a -10-extended promoter for TopoI and for RNAP is a novel observation that has not been reported for other bacteria. This is consistent with a role for TopoI in the formation/stability of the RNAP-DNA complex at the promoter in addition to transcript elongation. In my opinion, these are significant observations that should be of interest to researchers in transcription at the genome level. I have the following comments and suggestions for providing better understanding of the results and supporting the conclusions.

1. The basis of change in TopoI promoter occupancy sites in SCN treated cells remains unclear. SCN is a phenanthroline alkaloid that is expected to act as an DNA intercalator. DNA intercalation may affect DNA supercoiling and TopoI binding to chromosomal DNA loci. To further understand effect of SCN treatment on TopoI occupancy of different promoters, there should be a control experiment with a similar DNA intercalator that does not act as a strong TopoI inhibitor.

2. As the distance between ATG and transcription termination site increases with longer lengths of ORF, requirement of relieve of transcription driven Sc during elongation should take TopoI further away from the ATG. This should be analyzed to confirm that TopoI has a significant role in S. pneumoniae transcription elongation.

3. It is not known if the -10-extended promoter sequence motif is recognized specifically by TopoI, or if this sequence binds TopoI at high frequency because it is found upstream of highly transcribed divergent genes. The in vitro binding affinity of TopoI to this -10-extended sequence should be investigated.

4. The authors noted that residues in E. coli TopoI proposed for interaction with RNAP are not conserved in S. pneumoniae. It has been reported that M. tuberculosis TopoI utilized residues different from E. coli Topo I to interact with RNAP. https://pubmed.ncbi.nlm.nih.gov/28843989/ This evidence that different bacterial species may utilize distinct mechanism for direct interaction between TopoI and RNAP should be included in the Discussion.

5. On page 12 of Discussion, the authors proposed that “TopoI plays a role in head-on genes, likely resolving torsional stress generated when transcription and replication collide”. This proposed role is not clear and needs to be further explained, as positive Sc is generated ahead of both transcription and replication, and TopoI cannot relax positively supercoiled DNA.

Other minor comments:

i. In Author Summary “Binding of each of these enzymes to promoters is avoided by the specific inhibition of the other enzyme”, the meaning will be clearer if “avoided” is substituted with “prevented”.

ii. Page 4, “The increase in Sc mediated by SCN caused modulates the transcription of 10% of the genome” – The word “caused” should be removed.

iii. Page 5: “The twin supercoiled-domain model proposes that negative Sc domains are transiently generated, respectively behind and ahead of the moving RNAP” – This statement is incorrect; “and positive Sc” needs to be inserted after “negative Sc”.

iv. Page 9, “Identification of TopoI and RNAP peaka and identification of binding motifs” – “peaka” typo

v. Page 11, “RNAP-TopoI in vitro physical interaction has also been reported for its homologous in E. coli” - “homologous” should be replaced by “homologs”.

Reviewer #2: A very good paper defining topoisomerase I binding to the bacterial chromosome and which intriguingly indicates a direct interaction of Topo I with the RNA polymerase, and not only an overlap of its positioning on the chromosome with respect to transcriptional units. The paper may benefit for some revision with respect to the interpretation of the DNA binding motif of topo I.

Introduction

The introduction adequately covers the necessary background to bacterial topoisomerases but insufficient editing has unfortunately made it difficult to understand in places.

Some examples:

1. “In S. pneumoniae, the transcriptomic response to relaxation by inhibition of gyrase with novobiocin or by inhibition of TopoI by SCN, has revealed Sc-domains.” – Surely this is a mis-phrasing as inhibition of TopoI should prevent relaxation.

2. “The increase in Sc mediated by SCN caused modulates the transcription of 10% of the genome”

3. “AT-content correlates with UP domains.” Is this AT high or AT low content?

Major Comments

Results

P7 “The effect of SCN on the promoter occupancy by RNAP was practically negligible (compare Fig 2C and 2F), since most of the RNAP-enriched promoters remained enriched under SCN treatment (Fig 2F).”

(I presume this is actually “2E and 2F”?) Whilst the number of RNAP occupied promoters remained similar after SCN treatment 25% of the genes changed – this is hardly negligible. Did it affect the proportions of the enriched groups?

It is unclear as to whether the specificity of the TopoI binding motifs has been determined. We are told that Motif 2 is a -10 extended sequence and is “commonly found at pneumococcal promoters “ – as Motif 2 is only found at 18% of TopoI peaks how common is this Motif at promoter sequences where TopoI does not show enhanced binding?

Discussion

“Our ChiP experiments showed that RNAP preferentially located at promoter regions (Fig 2A) of highly expressed genes, and that promoters bound by both TopoI and RNAP had higher levels of expression than those bound by only RNAP (Fig 2A). These results indicate that TopoI plays a role in the activation of transcription.”

It is not immediately obvious to me why this supports TopoI being directly involved in the activation of transcription rather than it just being recruited more often at these sites where it is required more because transcription is higher.

Also, in work in Mycobacteria the TopoI was shown by ChiP to bind relatively evenly across the whole genome (Rani et al NAR 2019) but that topoI only showed peaks of cleavage activity at the highly expressed promoters – which is different to the finding of Ahmed also in mycobacteria (Ahmed PLoS Genetics 2017) shows simila binding of topoI and gyrase. Is there any suggestion for why this should be different in the Pneumococcus?

“In addition, structure-based modelling of RNAP-TopoI interaction in E. coli has identified the residues of TopoI and subunit RNAP B' [31] involved in the interaction. However, none these residues are conserved in the pneumococcal proteins. More structural studies in pneumococcus are needed to characterize this interaction.”

Whilst I appreciate that the residues identified in the E. coli RNAP-TopoI interaction are not present in Pneumococcus it is surprising that you have not commented on the fact that Rifampicin, which in your data (Fig 1) inhibits the TopoI-RNAP interaction, does so by binding the RNAP β subunit (which is also the site of the interaction shown in E. coli). Could this simply be the mechanism of Rifampicin interference with RNAP-TopoI binding?

In the discussion of the DNA binding motifs for the pneumococcal topoI the paper would significantly benefit from discussing the papers defining the target motifs for pneumococcal topoIV and gyrase (Leo et al 2005 JBC, Laponogov et al., 2009 NatStucBiol and 2010 PLoS One). Also, Sutormin NAR 2019 could be a good contribution to the discussion of the DNA motifs, even if this is E. coli, but the concept of a larger (130 bp) target motifs is interesting to include in the discussion.

The link to data in NCBI does not seem to find the data. (Though this could be NCBI!)

Minor comments and typos

The manuscript would be improved immensely by some additional editing work. The abstract and summary in particular appear to have been cut to fit the word limit without enough checking for continued clarity.

There are quite a lot of typos – this is not an exclusive list.

Introduction

Topoisomerase I is shown abbreviated to both “Topo I” and “TopoI”

Methods

P13 & P14: erythromycin not “erytromycin”

P14: “02%”

P15: “11% [vol/vol] formaldehyde” – vol/vol isn’t an informative without the original formaldehyde concentration e.g. 11% [vol/vol] formaldehyde (37% weight/vol)

Discussion

P10 “located close to and upstream RNAP” to “located close to and upstream of RNAP”

P11 “its homologous in E. coli” to “its homologue in E. coli”

Reviewer #3: In this study the authors have described the direct interaction of TopoisomeraseI and RNA Polymerase( RNAP) by SPR and then followed it up with ChIP-seq to determine the binding of these enzymes at the genomic sites in S. pneumoniae. Based on RNAP’s interaction with the topoisomerase and the co-localization of their peaks at the promoter regions, they conclude that Topoisomerase is involved in the formation and/or stability of the promoter - RNAP complex. This is an important work in the context of S. pneumoniae in understanding interdependence between transcription and topology as they have shown earlier that supercoiling homeostasis in this organism relies on transcriptional regulation of Topoisomerase I and to a less extent DNA gyrase. However, additional experiments are necessary to substantiate their present claims. The manuscript suffers from inadequate confirmatory experiments and also very difficult to read as it lacks in clarity in the statements on crucial data supporting their hypothesis. Too many sentence and typographical errors do not help their cause either.

1. Direct role of TopoisomeraseI in the activation of transcription is claimed based on a single experiment on RNAP interaction with the topoisomerase and co-localization of both the enzymes at the promoter region. Interaction should have been studied at least with yet another technique. If SPR assays were conducted with immobilized DNA and then passing the two proteins sequentially would have been a more insightful experiment to see their direct interaction on DNA . In this experiment, one can perform order of addition experiment to understand the interaction more meaningfully. Which sub -unit of RNAP is interacting with TopoisomeraseI ? A large number of studies have been published on RNAP sub unit interaction with various proteins using a whole range of techniques including SPR. Also, the experiment described does not provide information on what fraction of TopoisomeraseI and RNAP interact with each other in the cell. Is it a small fraction or large fraction would be important to know. As they have high quality antibodies (see later point), it is easy to estimate the extent of interaction.

2. Topoisomerase I binding to the promoter does not suggest that it activates RNAP. To test the activation model, increase in complex formation or stability of the complex as they claim, the authors need to check the influence of Topoisomerase I on RNAP closed complex, open complex formation and if necessary(if the enzyme has no effect on the earlier steps) promoter clearance and abortive initiation assays. To test the effect of the enzyme on RNAP –promoter complex stability, complex decay experiments need to be performed. Without these assays, the activation model is a conjecture.

3. Both Topoisomerase I and RNAP binding to the same site at the promoter is problematic. If indeed both are recruited to the same site, they would compete out each other instead of cooperating for transcription. First, one can test and validate binding of both the enzymes to the extended – 10 containing promoter sequences by EMSA to confirm the consensus site they derived from ChiP Seq. If indeed they are bound, one has to displace the other! RNAP binds a large region at the promoters and it would be interesting to know where Topoisomerase I binds.- upstream , downstream or on RNAP.

4. Figures- Fig1A- Purified Topoisomerase I has at least 10 additional proteins. If silver stained, more bands will be seen. Similarly , RNAP preparation has 5 additional bands, some of which are of lower intensity. This is problematic as in the described SPR amino coupling is done and some of these proteins may interfere resulting in erroneous interpretation of the data. Fig1B is very poor representation of RNAP. With the availability of crystal structure of RNAP from various species of bacteria, one can draw a better, a more representative cartoon of the enzyme. As such, I do not see the need for the figure as it is not serving any purpose. If TopoisomeraseI interaction was shown with a subunit of RNAP, perhaps, the cartoon would serve some purpose. Supplementary figure: Antibody purification is described in detail. There is no ‘High’ chain in an antibody. Replace with ‘heavy’chain.

5. ChiP Seq experiments : Some of the data are clear whereas others beg some explanation. Some others are difficult to explain. For example, why treatment with Topoisomerase I inhibitor SCN should alter the RNAP occupancy? If TopoI activity is necessary for transcription activation, the inhibitor (SCN) treatment should alter RNA transcripts levels of specific set of genes. Further, I am not able to understand Rifampicin affecting RNAP occupancy as its action is after the formation of the ternary complex and early step of elongation(authors have written and cited this information).

Additional points/corrections:

Fig 1D – Why SCN or Rifampicin addition should change RUs?

Many sentence and typographical errors-

Author summary – sentence 2 and others.

Introduction: ...caused...modulates...; Relaxing topoisomerase?

....generated respectively, behind and ahead of the moving RNAP

...which, otherwise, would difficult transcription elongation

Results: SCN dropped the vale( ??) to 40.9 units

Many others including a sub title in the Results section – ‘ Identification of TopoI and RNAP peaka and identification of binding motifs’

**Have all data underlying the figures and results presented in the manuscript been provided?**

Reviewer #1: **No: **No data is available through the links provided https://dataview.ncbi.nlm.nih.gov/object/PRJNA656439?reviewer=hrj007r86ulcrnktia18

http://microbes.ucsc.edu/cgibin/hgTracks?hgS_doOtherUser=submit&hgS_otherUserName=Pablo_Hernandez&hg

S_otherUserSessionName=S_pneumoniae_ChIP-seq

I assume that this is a technical issue, and data will become available

Reviewer #2: Yes

Reviewer #3: Yes

PLOS authors have the option to publish the peer review history of their article (what does this mean?). If published, this will include your full peer review and any attached files.

Reviewer #1: No

Reviewer #2: No

Reviewer #3: No

---

## [Editor Report · Decision Letter 1]

10 Apr 2021

Dear Adela,

I am pleased to inform you that your manuscript entitled "Genome-wide proximity between RNA polymerase and DNA topoisomerase I supports transcription in Streptococcus pneumoniae" has been editorially accepted for publication in PLOS Genetics. Congratulations!

Yours sincerely,

Josep Casadesús

Section Editor: Prokaryotic Genetics

PLOS Genetics

Comments from the reviewers (if applicable):

**Data Deposition**

http://datadryad.org/submit?journalID=pgenetics&manu=PGENETICS-D-20-01865R1

**Press Queries**

---

## [Editor Report · Acceptance letter]

26 Apr 2021

PGENETICS-D-20-01865R1 

Genome-wide proximity between RNA polymerase and DNA topoisomerase I supports transcription in Streptococcus pneumoniae 

Dear Dr de la Campa, 

We are pleased to inform you that your manuscript entitled "Genome-wide proximity between RNA polymerase and DNA topoisomerase I supports transcription in Streptococcus pneumoniae" has been formally accepted for publication in PLOS Genetics! Your manuscript is now with our production department and you will be notified of the publication date in due course.

With kind regards,

Katalin Szabo

PLOS Genetics

On behalf of:
